# Hybrid Collagen Hydrogel/Chondroitin-4-Sulphate Fortified with Dermal Fibroblast Conditioned Medium for Skin Therapeutic Application

**DOI:** 10.3390/polym13040508

**Published:** 2021-02-08

**Authors:** Manira Maarof, Masrina Mohd Nadzir, Lau Sin Mun, Mh Busra Fauzi, Shiplu Roy Chowdhury, Ruszymah Bt Hj Idrus, Yogeswaran Lokanathan

**Affiliations:** 1Centre for Tissue Engineering and Regenerative Medicine, Faculty of Medicine, Universiti Kebangsaan Malaysia, Jalan Yaacob Latif, Cheras, Kuala Lumpur 56000, Malaysia; manira@ppukm.ukm.edu.my (M.M.); fauzibusra@ukm.edu.my (M.B.F.); shiplu56@gmail.com (S.R.C.); ruszyidrus@gmail.com (R.B.H.I.); 2School of Chemical Engineering, Engineering Campus, Universiti Sains Malaysia, Nibong Tebal, Pulau Pinang 14300, Malaysia; chmasrina@usm.my (M.M.N.); sinmun3587@hotmail.com (L.S.M.); 3Department of Physiology, Faculty of Medicine, Universiti Kebangsaan Malaysia, Jalan Yaacob Latif, Cheras, Kuala Lumpur 56000, Malaysia

**Keywords:** secreted proteins, dermal fibroblast conditioned medium, collagen, skin substitute

## Abstract

The current strategy for rapid wound healing treatment involves combining a biomaterial and cell-secreted proteins or biomolecules. This study was aimed at characterizing 3-dimensional (3D) collagen hydrogels fortified with dermal fibroblast-conditioned medium (DFCM) as a readily available acellular skin substitute. Confluent fibroblasts were cultured with serum-free keratinocyte-specific medium (KM1 and KM2) and fibroblast-specific medium (FM) to obtain DFCM. Subsequently, the DFCM was mixed with collagen (Col) hydrogel and chondroitin-4-sulphate (C4S) to fabricate 3D constructs termed Col/C4S/DFCM-KM1, Col/C4S/DFCM-KM2, and Col/C4S/DFCM-FM. The constructs successfully formed soft, semi-solid and translucent hydrogels within 1 h of incubation at 37 °C with strength of <2.5 Newton (N). The Col/C4S/DFCM demonstrated significantly lower turbidity compared to the control groups. The Col/C4S/DFCM also showed a lower percentage of porosity (KM1: 35.15 ± 9.76%; KM2: 6.85 ± 1.60%; FM: 14.14 ± 7.65%) compared to the Col (105.14 ± 11.87%) and Col/C4S (143.44 ± 27.72%) constructs. There were no changes in both swelling and degradation among all constructs. Fourier transform infrared spectrometry showed that all groups consisted of oxygen–hydrogen bonds (O-H) and amide I, II, and III. In conclusion, the Col/C4S/DFCM constructs maintain the characteristics of native collagen and can synergistically deliver essential biomolecules for future use in skin therapeutic applications.

## 1. Introduction

Skin substitutes are important as an alternative to split skin grafts (SSG), especially for non-healing chronic wounds, deeper and larger wounds, or for non-healing smaller wounds due to underlying factors such as infection, ischemia, immunosuppression, and metabolic conditions [1]. During wound healing, secretory mediators or growth factors are essential for activating or inhibiting the signaling pathways required for wound healing. Therefore, these mediators or growth factors have the potential to be used as supplementary therapies in wound treatment. Single-layer keratinocytes, single-layer fibroblasts, and bilayered skin constructs have healing potential [2,3,4] for tissue regeneration. These constructs secrete essential factors such as cytokines, chemokines, and growth factors that are important for wound healing [5].

Fibroblasts are well-known sources for secreting these essential factors, which can be collected in the form of dermal fibroblast conditioned medium (DFCM). Previously, we reported the identification of various wound healing mediators such as fibronectin, serotransferrin, serpin, and collagen, which are secreted abundantly by human dermal fibroblasts (HDFs), in DFCM [6]. These secreted mediators are under the extracellular matrix (ECM) or cytoskeleton, signaling a molecule or calcium-binding protein class that is involved in molecular- and biological-level cellular processes and wound healing [6]. The supplementation of DFCM into 2-dimensional (2D) keratinocyte culture and scratch assay increased human epidermal keratinocyte (HEK) attachment, proliferation, and migration [7].

Readily available acellular skin substitute production using DFCM is crucial and should incorporate a biocompatible material to form a 3D structure providing optimum stability and efficient DFCM controlled release. Scaffolds or biomaterials are the main components of skin substitutes and are derived from various sources, chemical components, and applications [8,9]. Biomaterials can be classified into two main groups: natural (e.g., biological products derived from animal, plant, fungal, or bacteria sources) or synthetic (e.g., organic or inorganic polymers) [9]. Naturally derived biomaterials, which include collagen, gelatin, silk, cellulose, and chitin/chitosan, are the preferred choices compared to synthetic biomaterials for repairing or replacing tissues or organs [10]. The most likely reasons for this are their advantages in terms of biocompatibility, biodegradability, remodeling, and essential biological functions [11,12,13]. Besides the indispensable factors, which include their structural, morphological, biological, and mechanical properties, the fabricated biomaterials are also crucial for producing a functional skin substitute to mimic the native skin structure [14,15].

Previously, we successfully extracted collagen from ovine tendon and fabricated scaffolds in the form of sponge, hydrogel, and film to be used for developing tissue substitutes for clinical application and for in vitro 3D models [16,17,18]. The physicochemical and mechanical properties of natural collagen can be enhanced with the addition of chondroitin-4-sulphate (C4S). C4S is a glycosaminoglycan (GAG) chain that consists of repeating disaccharide units β-1,3-linked *N*-acetyl galactosamine (GalNAc) and β-1,4-linked D-glucuronic acid (GlcA) [19]. C4S is widely used in incorporation with biomaterials, as it provides compression resistance and promotes cell proliferation for tissue repair [20,21,22]. C4S has also been used extensively in the drug delivery and tissue engineering fields because of its properties, i.e., it is a major ECM component, and is non-poisonous, biodegradable, and biocompatible [23]. C4S increases the amount of collagen incorporated into fibrils for forming collagen self-assembly by affecting nucleation during the lag period [24]. C4S can reduce enzyme activity and prolong gelling time [25].

Here, we aimed to fabricate and characterize a 3D acellular collagen hydrogel fortified with C4S, and different types of DFCM, to compare its potential with the intended acceptable porosity, protein permeability, and mechanical strength as a readily available acellular skin substitute for skin therapeutic applications. The characterization will demonstrate the future potential of the fabricated ovine collagen hydrogel as a delivery vehicle for drug and supplement delivery for skin tissue regeneration.

## 2. Materials and Methods

### 2.1. Cell Isolation and Culture

Redundant skin tissue samples were obtained from three consenting healthy patients (n = 3) who had undergone abdominoplasty or face-lift surgery. The tissue samples were processed and cultured as described elsewhere [3]. Briefly, the redundant skin was minced, digested with 0.6% type I collagenase (Worthington, Columbia, NJ, USA) for 4–6 h in a 37 °C incubator shaker (Stuart, Staffordshire, UK) and dissociated using 0.05% trypsin-EDTA (Gibco, Gaithersburg, MD, USA) for 8–10 min. The digested cells were then suspended in a co-culture medium (equivalent mixture (1:1 ratio) of HEK growth medium, i.e., EpiLife™ (Gibco) and HDF growth medium, i.e., F-12:Dulbecco’s modified Eagle’s medium (F12:DMEM; Sigma-Aldrich, St. Louis, MO, USA) and supplemented with 10% fetal bovine serum (FBS; Gibco)). The cells were seeded into six-well culture plates (Greiner Bio-One, Monroe, NC, USA) at 37 °C in 5% CO_2_, and the medium was replaced every 2–3 days. The HDFs were sub-cultured in a T75 flask (Nunc, Rochester, NY, USA) in F12:DMEM + 10% FBS until passage 3 (P3).

### 2.2. DFCM Preparation and Collection

DFCM was prepared using 80–100% confluent P3 fibroblasts as described previously [6]. Briefly, the medium was removed, and the cells were washed twice with Dulbecco’s phosphate-buffered saline (DPBS, Sigma-Aldrich, St. Louis, MO, USA). Then, fresh serum-free HEK growth medium (EpiLife; Gibco; referred to as KM1), defined keratinocyte serum-free medium with supplement (DKSFM; Gibco; referred to as KM2), or HDF culture medium (F12:DMEM without serum; Sigma-Aldrich; referred to as FM) was added to the HDF separately. The cells were incubated at 37 °C in a 5% CO_2_ incubator for 72 h, and the waste medium was collected as DFCM-KM1, DFCM-KM2, and DFCM-FM, respectively. The DFCM was filtered using a 3 kDa Amicon Ultra-15 centrifugal filter (Merck Millipore, Burlington, MA, USA) to concentrate the proteins. The protein concentration was determined using a bicinchoninic acid (BCA) assay. The absorbance was measured at 562 nm (PowerWave XS, BioTek, Winooski, VT, USA), and the protein concentration was calculated by comparing it with a known protein standard (Sigma-Aldrich). The samples were stored at −80 °C for further analysis and experiments. The DFCM concentrations used for scaffold fabrication with collagen and C4S were 200 μg/mL (DFCM-KM1 and DFCM-KM2) and 400 μg/mL (DFCM-FM). The DFCM concentrations were determined on the basis of the effective DFCM concentrations for cell attachment, proliferation, and migration reported in a previous in vitro study [26].

### 2.3. Fabrication of the Collagen Hydrogel with C4S and DFCM

Collagen hydrogel was produced with in-house-prepared type I collagen (Col I) extracted from ovine tendon as previously described [16]. The collagen in 0.35 M acetic acid solution (0.6% (*w*/*v*)) was mixed with 2.4% (*w*/*v*) C4S (Sigma-Aldrich) in an ice bath with a magnetic stirrer until homogenized, and was neutralized by adding 1 M sodium hydroxide (NaOH; Sigma-Aldrich) to pH 7. Then, the collagen mixture was centrifuged (ScanSpeed 1248R, LaboGene, Bjarkesvej, Lillerød, Denmark) at 4000 rpm at 4 °C for 2 min to remove air bubbles and to completely blend with the different DFCM. The mixture was then incubated at 37 °C to initiate gelation to form the 3D constructs, which were termed Col/C4S/DFCM-KM1, Col/C4S/DFCM-KM2, and Col/C4S/DFCM-FM (n = 3). Collagen alone (Col) and collagen with C4S (Col/C4S) were used as the control. The gross morphology and microstructure of the constructs were observed via scanning electron microscopy (SEM; Quanta FEG 450, FEI; Eindhoven, North Brabant, The Netherlands).

### 2.4. Turbidity of the Collagen Hydrogel Constructs

The turbidity of the constructs was measured using a Cary 60 UV–Vis spectrophotometer (Agilent, Santa Clara, CA, USA). The Col/C4S/DFCM constructs were prepared as described above and were immediately poured into 1.5 mL cuvettes after centrifuging at 4 °C. The construct was incubated at 37 °C, and the turbidity was measured at 15 min intervals for 1 h at 310 nm.

### 2.5. Fourier Transform Infrared Spectrometry (FTIR)

The chemical structure of the constructs was characterized using FTIR (IR Prestige-21, Shimadzu, Nakagyo-ku, Kyoto, Japan) through functional group identification. The constructs were prepared as described above, and the FTIR spectra of the constructs (n = 3) were recorded in the frequency range of 600–4000 cm^−1^. The data were analyzed using Shimadzu IR Solution FTIR (spectroscopy) software (Shimadzu).

### 2.6. Porosity of the Collagen Hydrogel Constructs

The porosity of the constructs was measured with the liquid replacement method [27]. All constructs (n = 3) were freeze-dried overnight using a freeze dryer (Ilshin, Siheung-si, Gyeonggi-do, Korea), and the initial dried-to-constant weight (*W_i_*) was measured. The constructs were then immersed in absolute ethanol, and their weight, in a known volume of ethanol, was measured as volume (*V)*. The ethanol-immersed constructs were agitated for 15 min using a sonicator (Fisherbrand, Göteborg, Sweden). Then, the constructs were removed, blotted with filter paper to remove the excess ethanol, and weighed (*W_f_*) immediately. The porosity (%) was calculated using the equation below, where *V* is the volume of the construct and *p* is the density of absolute ethanol (0.789 g/mL):$$\mathrm{Porosity}\;\left(\%\right)=\frac{\left(W_{f}-W_{i}\right)\times 100}{Vp}$$

### 2.7. Swelling Analysis

The Col/C4S/DFCM constructs (n = 3) were freeze-dried overnight using a freeze dryer (Ilshin), and the percentage of swelling was determined by immersing the dried collagen constructs in 10× DPBS at room temperature. The initial weight (*W_d_*) was measured using a weighing balance before the constructs were immersed in 10× DPBS (Sigma-Aldrich). At every 15 min up to 2 h, the constructs were removed, blotted with filter paper to remove excess water, and immediately weighed to obtain the wet weight (*W_w_*). The construct swelling (%) was calculated using the following equation:$$\mathrm{Swelling}\;\left(\%\right)=\frac{\left(W_{w}-W_{d}\right)\times 100}{W_{d}}$$

### 2.8. Mechanical Strength Testing

The strength of the constructs (n = 3) was determined using the Bloom strength test according to the International Organization for Standardization standard ISO 9665:1998, which is the common method for measuring the strength of soft gels. The test was performed using a TA.XTplus Texture Analyzer (Stable Micro Systems, Godalming, Surrey, UK). The constructs were placed at the center of the analyzer with the probe 0.5 inches above the construct surface. The probe penetrated the constructs to a target distance or depth of 2 mm at the speed of 0.5 mm/s and contact force of 5 g, and then retracted. The maximum peak force generated was the strength of the constructs. The results were averaged for three independent runs.

### 2.9. Degradation of the Collagen Hydrogel Constructs

The constructs were prepared (n = 3 per sample) in Transwell cell culture inserts (Greiner Bio-One, Kremsmünster, Austria) and incubated in 0.0015% type I collagenase (2 U/mL) (Whartington, UK) at 37 °C for 24 h according to the protocol described by Sakamoto et al. [28]. A total 200 µL of the released protein in collagenase solution was sampled for the first 30 min and at subsequent 2 h intervals for 24 h (n = 3 each time). The solutions were kept at −80 °C until they were measured using the BCA assay, with the absorbance read at 562 nm.

### 2.10. Statistical Analysis

The quantitative results are shown as the mean ± standard deviation (SD). The results were analyzed with analysis of variance (ANOVA), and the difference between groups was significant if *p* < 0.05.

## 3. Results

### 3.1. Morphology and Turbidity of the Collagen Hydrogel Constructs

The gross observation of the polymerized Col/C4S/DFCM constructs showed that they were soft, semi-solid, translucent, and more turbid when mixed with C4S (Figure 1A). The turbidity changes in the constructs were measured by the absorbance (au) at 310 nm. The Col/C4S construct became slightly cloudy and more turbid compared to the other constructs, and the construct turbidity increased drastically after incubation at 37 °C, and achieved a steady-state after 15 min incubation (Figure 1B). However, the DFCM constructs showed lower turbidity compared to the non-DFCM constructs regardless of the type of DFCM incorporated into the construct. All constructs successfully formed a gel within 1 h incubation at 37 °C.

### 3.2. Chemical Characterisation of the Collagen Hydrogel Constructs

The IR spectra showed peak absorbance that represented the chemical composition of the constructs (Figure 2). All groups showed similar peak patterns, which consisted of a strong functional group oxygen–hydrogen bond (O-H) at 3330–3340 cm^−1^. Peaks were also observed at 1639–1640, 1500–1600 and 1260–1300 cm^−^^1^, which represented the wavelengths of amide bands I, II, and III, respectively. Amide I, a major band in collagen, mainly from the carbon–oxygen double bond (C=O), was found in the 1600–1700 cm^−1^ range, whereas amide II from the nitrogen–hydrogen bond (N-H) was found in the 1500–1600 cm^−1^ range, and amide III from the carbon–nitrogen (C-N), or carbon–oxygen single bond (C-O), was found in the 1200–1300 cm^−1^ range. This confirms that all constructs maintained the primary collagen composition, which mainly consists of carbon (C), nitrogen (N), and oxygen (O) elements. All constructs also had peaks at ranges of 2100–2260 cm^−1^ and 2200–2300 cm^−1^, which represent the wavelengths for the functional groups carbon–carbon triple bond of organic molecule alkynes and carbon–nitrogen triple bond of nitrile groups that may be derived from salt. Table 1 lists the pH of the constructs.

### 3.3. Collagen Hydrogel Construct Microstructure

The surface of the constructs was assessed at 5000× and 10,000× magnification under SEM. Most of the constructs showed collagen fibril aggregates, resulting in thicker collagen fibrils. The homogeneously overlapping collagen fibrils observed in Col/C4S/DFCM-FM had fewer collagen fibrils arranged in bundles (Figure 3). However, all constructs showed a densely packed microstructure, and observation of the collagen fibrils under SEM was not distinct, possibly due to the crystalline phase during the freeze-drying closing the pores of the constructs.

### 3.4. Physical Retention Characteristics of the Collagen Hydrogel Constructs

The dry weight of the constructs was 0.011–0.039 g (Figure 4A). The percentage of porosity of the Col/C4S construct (93.3% ± 8.1) was higher compared to that of the DFCM constructs (Col/C4S/DFCM-KM1: 43.8% ± 25.4; Col/C4S/DFCM-KM2: 35.2% ± 9.8; Col/C4S/DFCM-FM: 14.1% ± 7.7). Besides, the porosity of the Col construct (71.8% ± 13.8) was significantly higher compared to that of the Col/C4S/DFCM-FM construct (Figure 4B). All Col/C4S/DFCM constructs showed lower porosity, although all of the constructs had been fabricated with the same concentration of Col/C4S. All constructs showed a similar swelling pattern: a continuous increase in swelling percentage throughout the 2 h immersion in DPBS (Figure 4C).

### 3.5. Mechanical Strength of the Collagen Hydrogel Constructs

All constructs had a force of <2.5 N, which indicates that they are soft collagen hydrogels (Figure 5A). The force of the Col construct (0.88 ± 0.0009 N) and Col/C4S construct (0.70 ± 0.1478 N) was slightly lower compared to that of the Col/C4S/DFCM constructs. However, there were no significant differences between all groups.

### 3.6. Protein Release of the Collagen Hydrogel Constructs

Most of the constructs were fully digested by type I collagenase (0.0015%) after 24 h incubation at 37 °C (Figure 5B). All constructs showed the same protein release pattern, whereby the protein concentrations increased with incubation time and ranged 200–250 µg/mL after the 24 h incubation.

## 4. Discussion

The current strategy in wound healing management involves cell-based therapy, an acellular-based approach and standard wound dressing. Skin substitutes are a standard alternative treatment using tissue engineering technology that protect against microorganism invasion, control the loss of water vapor, act as a wound cover, provide cytokines and growth factors to enhance wound healing, and provide a supportive structure to the wound area [29,30]. Efforts had begun years ago to avoid the drawback of longer waiting time in producing sufficient cell numbers for skin substitutes. Therefore, the use of conditioned medium is an attractive strategy due to its advantages of production in large amounts from allogenic sources and high protein content that promotes cell growth, cell differentiation, and tissue repair [31]. DFCM improves cell proliferation and migration rates, and also contains various mediators, including ECM, growth factors, chemokines and cytokines, which are involved in wound healing and the regeneration of skin cells [6,32,33]. In the present study, we explored the fabrication of collagen hydrogel constructs incorporating C4S and DFCM, and characterized the constructs for future use as acellular skin substitutes. The hydrogel scaffold is widely used in tissue engineering because of its structural similarity to most ECM tissues [34].

Collagen is the main component of the ECM and is the most common biomaterial used because of its low immunogenicity, biocompatibility, biodegradability, hydrophilicity, and availability [9,35,36]. Here, we tested the mechanical strength of the collagen hydrogel constructs using the Bloom strength test, where the hardness or strength of the constructs was measured using the peak force (in N) or maximum value for a single compression cycle obtained [37]. Higher force peaks represent higher collagen strengths. In the present study, we successfully fabricated a collagen hydrogel with a soft, tissue-like texture via the incorporation of C4S and DFCM, as all constructs had a force of <2.5 N (Figure 1A and Figure 5A). The Col and Col/C4S constructs had slightly lower force compared to the Col/C4S/DFCM constructs. This suggests that DFCM might have an important role in the strength properties of the constructs, as others have reported that incorporating growth factors in the scaffold promoted interaction between the hydrogel scaffold structure and mechanical properties [38].

All constructs mainly consisted of carbon, nitrogen, and oxygen elements with the functional groups amide I, II, and III and a hydroxyl group (Figure 2), which proves that the DFCM and non-DFCM constructs maintain the chemical characteristics of collagen without modification [39,40]. The C4S-containing construct had a more turbid or opaque appearance, probably because of the increased fibril formation [41]. The changes in the organization of the collagen network, known as fibrillogenesis, occur with the presence of C4S, which induces collagen fibril bundling and thickening [42,43]. The turbidity analysis (Figure 1B) showed the same pattern as that in a previous study and involved three phases: (a) lag phase, i.e., the turbidity does not change; (b) growth phase, i.e., turbidity decreases rapidly; and (c) maturity, i.e., turbidity is stable [44]. However, the DFCM-containing constructs showed less turbidity compared to the Col and Col/C4S constructs even though all constructs contained the same concentration of C4S. This is because the changes in pH to slightly alkaline conditions as well as the presence of salts in the DFCM itself might have affected the fibril formation [45]. However, the collagen fibrils for all constructs could not be observed clearly, which might have been due to the crystalline phase during freeze-drying closing the pores of the constructs or because of the presence of a small amount of salts in the DFCM, which usually tends to hydrolyze in water and form discrete nanoparticles [46].

Collagen in the form of hydrogel is suitable for various applications, including biomedical and tissue engineering, as it can absorb water molecules or swell when in contact with aqueous solutions due to the presence of hydrophilic groups in its backbone [47,48]. This would enable the diffusion of oxygen, proteins, and growth factors that are important in drug delivery, cell encapsulation, wound dressing, and tissue repair [47,49,50]. Swelling is one of the important parameters for evaluating the structure of a biomaterial. Here, all constructs showed an increasing percentage of swelling with time (Figure 4C) due to the ability of the collagen hydrogel to swell. In addition, the C4S content contains more hydrophilic groups, which in turn enhances water-uptake capacity, attracting more water to be absorbed into the constructs [51,52,53]. The rate of swelling depends on several factors, including porosity and porous structure [49].

The porosity of matrices (60–90% porous) is important for facilitating cellular infiltration and tissue ingrowth [54]. In the present study, we analyzed the porosity of the dried constructs with ethanol because it easily penetrates into the pores of the constructs without causing construct shrinkage or swelling [55]. The porosity test showed that the C4S-containing construct had slightly higher porosity compared to the construct without C4S (Figure 4B), which was also reported by Nadzir et al. [52]. However, the DFCM-fortified constructs had lower porosity, which might have been due to crystalline phase during freeze-drying or because of the presence of salts in the DFCM. The culture medium used to prepare DFCM contains a certain amount of inorganic salts; calcium chloride might have filled the pores of the constructs, as shown by the SEM image in Figure 3. Therefore, the DFCM should be dialyzed before being used for fabrication to remove small components such as salt.

Degradation is the process of breaking down collagen into simpler compounds with different methods such as proteolysis, or thermal or chemical treatment, at an optimal rate [56]. The degradation or protein release data in Figure 5B show that all constructs were fully degraded within 8 h of incubation with type I collagenase, which indicates efficient degradation for sustained release of the proteins [57]. Another study has also shown that collagen constructs incubated at 37 °C and 5% CO_2_ for 24 h released 20–70% of the biomolecules encapsulated in the construct [58]. In the present study, the degradability properties of the constructs are expected to facilitate the sustained release of the encapsulated DFCM proteins to the target area to stimulate wound healing [57]. However, in certain cases, a faster rate of degradation may cause improper tissue regeneration, as the 3D structure of the skin constructs should be sustained for at least three weeks to allow cell growth and vascularization to take place [59,60].

Numerous commercially available skin substitutes have been introduced over the decade. However, the ideal skin substitute that matches the functional native skin still has not been found due to several integral factors that require research focus [61,62,63]. The common properties of skin substitutes are to prevent severe fluid loss but still allowing minimum water exchange, able to swell for nutrient and oxygen diffusion, non-toxic, non-inflammatory, non-immunogenic and also biodegradable to allow host tissue regeneration [62]. In the present study, we show that collagen hydrogels fabricated with C4S and DFCM can potentially be used as an acellular skin substitute due to its suitable swelling, degradation, chemical, and mechanical properties that mimic the common biomaterials used for skin substitutes. The collagen hydrogel has been proven suitable for loading DFCM and can slowly release it to enhance wound healing. Additionally, C4S enhances the mechanical strength of the hydrogel while promoting wound healing. However, further studies are needed to exploit the biocompatibility of this construct composite on cellular response and cytotoxicity in vitro and in vivo. Further, comprehensive optimization and assessment should be considered, which includes the DFCM loading capacity and release profile according to requirements, before this construct can be used in clinical applications.

## 5. Conclusions

Our findings show that the fabricated Col/C4S/DFCM hydrogel constructs maintain the characteristics of collagen. The swelling and degradation properties of the constructs are suitable for the diffusion of DFCM, and the construct could have potential as a readily available acellular skin substitute that acts as an alternative carrier for delivering essential mediators and growth factors that enhance wound healing. We will investigate the efficacy of this delivery system in enhancing skin regeneration and its therapeutic application in future studies.

## Figures and Tables

**Figure 1 polymers-13-00508-f001:**
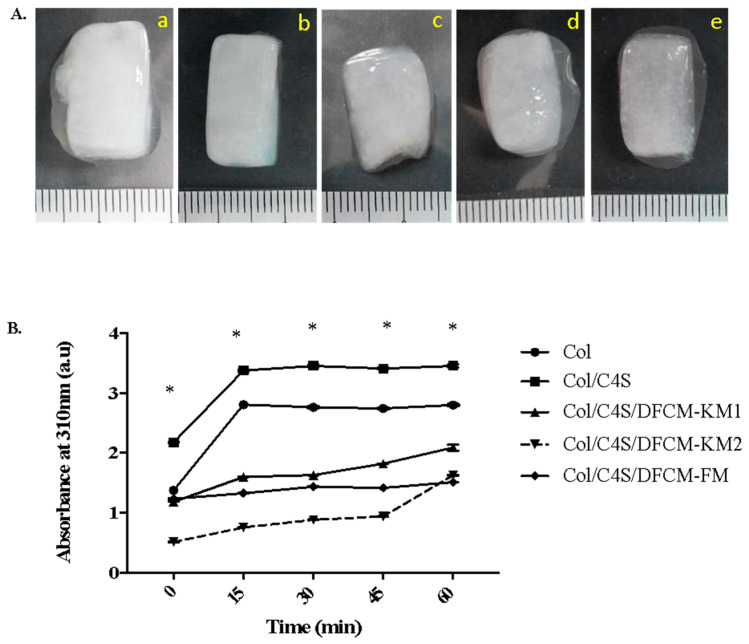
(**A**) The morphology of the polymerized constructs (a: Col; b: Col/C4S; c: Col/C4S/DFCM-KM1; d: Col/C4S/DFCM-KM2; e: Col/C4S/DFCM-FM). The constructs were soft, semi-solid and translucent, and more turbid when mixed with C4S. (**B**) Construct turbidity. Col/C4S construct became slightly cloudy and more turbid, and significantly more so compared to the other constructs (* significantly compared to other groups (*p* < 0.001)).

**Figure 2 polymers-13-00508-f002:**
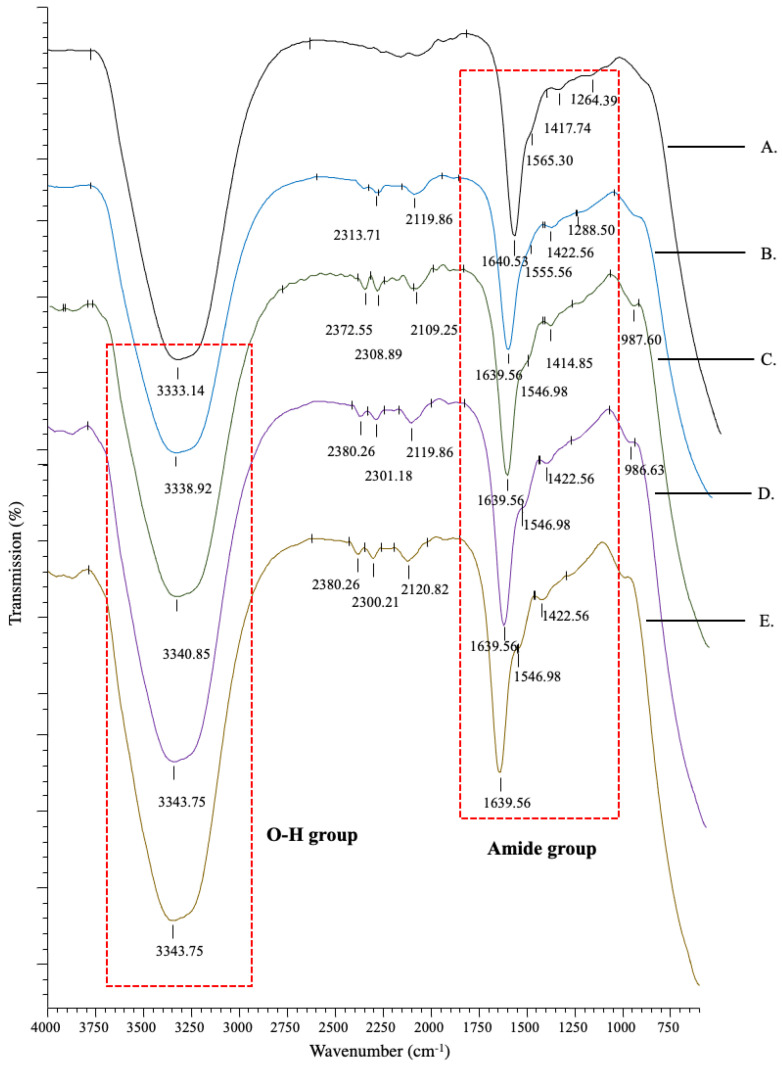
IR transmittance spectra of the constructs ((**A**): Col; (**B**): Col/C4S; (**C**): Col/C4S/DFCM-KM1; (**D**): Col/C4S/DFCM-KM2; (**E**): Col/C4S/DFCM-FM). All groups showed similar peak patterns, which consisted of the functional group O-H bond at 3330–3340 cm^−1^, and amide bands I, II, and III at wavelengths of 1639–1640, 1500–1600, and 1260–1300 cm^−1^, respectively.

**Figure 3 polymers-13-00508-f003:**
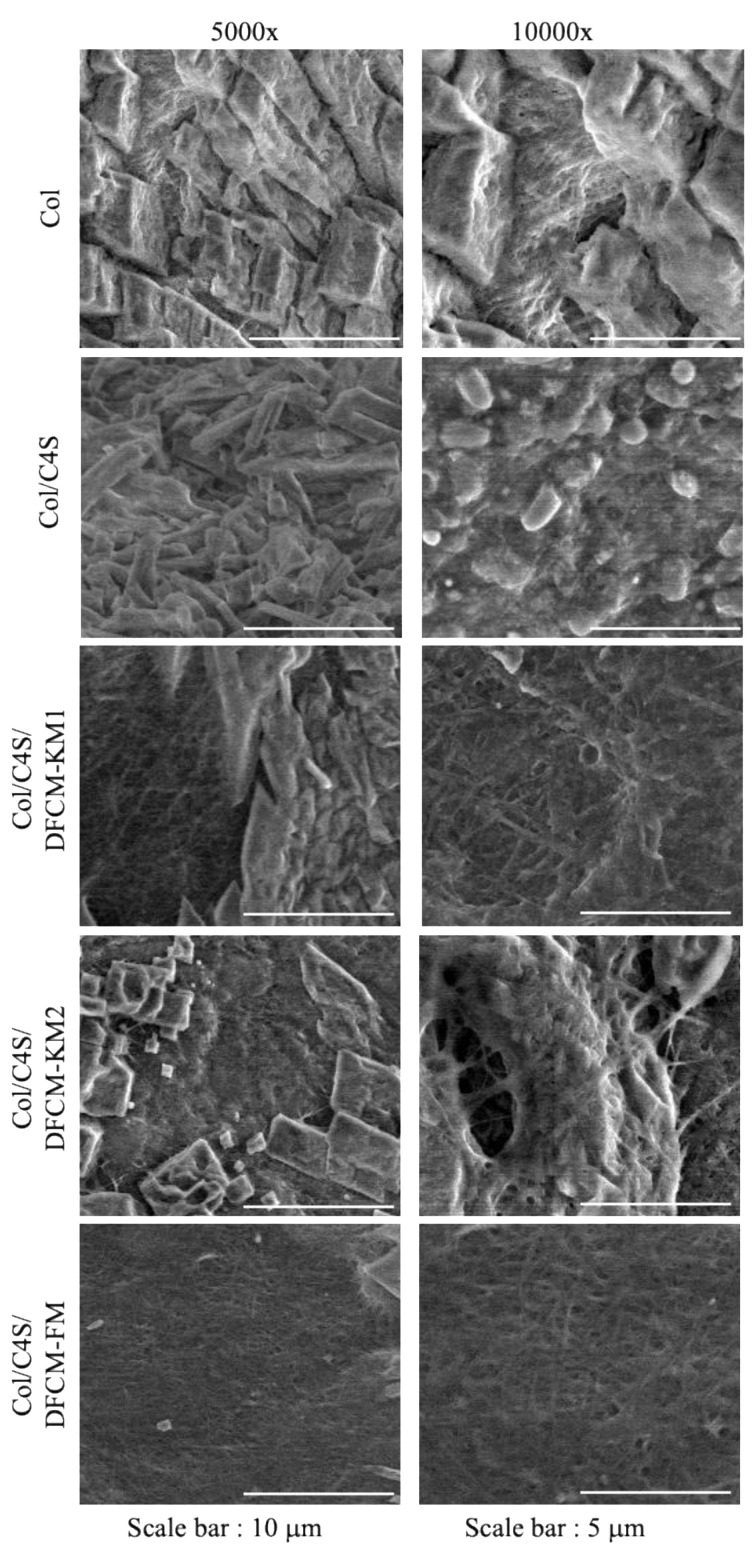
SEM images of Col/C4S/DFCM constructs at 5000× and 10,000× magnification. The collagen fibrils were not clearly observed under SEM analysis.

**Figure 4 polymers-13-00508-f004:**
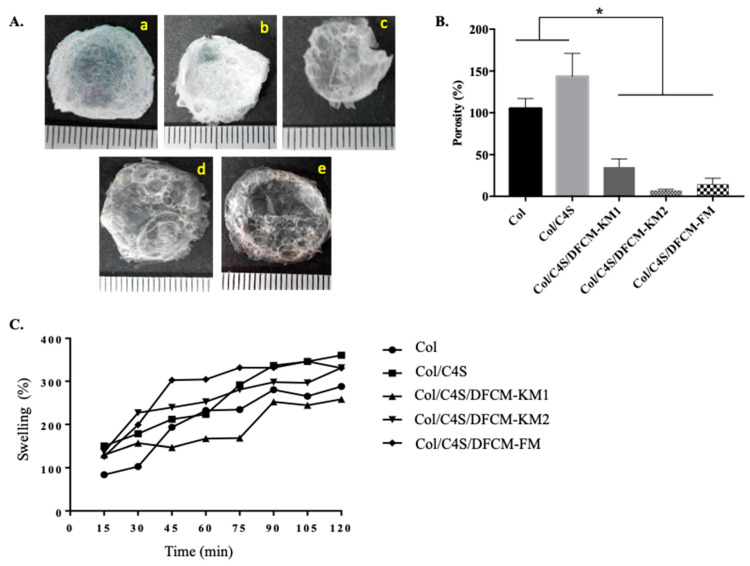
(**A**) The morphology of the freeze-dried constructs (a: Col; b: Col/C4S; c: Col/C4S/DFCM-KM1; d: Col/C4S/DFCM-KM2; e: Col/C4S/DFCM-FM). (**B**) The porosity of the constructs. The Col and Col/C4S constructs showed significantly higher (*) porosity compared to the Col/C4S/DFCM constructs. (**C**) The percentage of swelling of the constructs. Most of the constructs showed similar swelling patterns.

**Figure 5 polymers-13-00508-f005:**
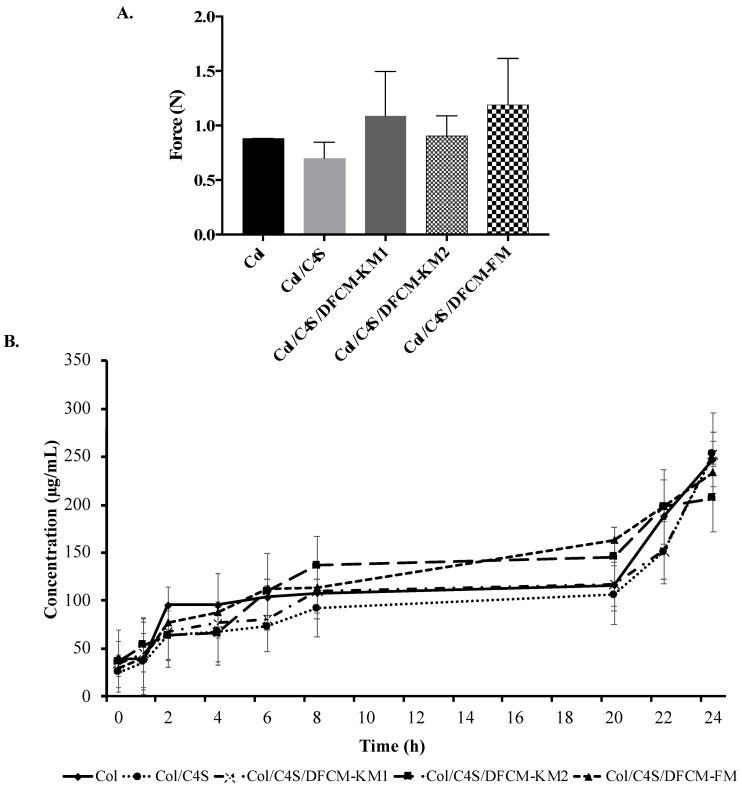
(**A**) Mechanical strength of the constructs. All constructs had force of <2.5 N, indicating that they were soft collagen hydrogel constructs. (**B**) Protein release from the constructs showing the same release pattern, i.e., the protein concentrations increased with incubation time.

**Table 1 polymers-13-00508-t001:** pH of the constructs.

Collagen Construct	pH
Col	7.15
Col/C4S	7.16
Col/C4S/DFCM-KM1	7.56
Col/C4S/DFCM-KM2	7.54
Col/C4S/DFCM-FM	7.47

## Data Availability

The data presented in this study are available on request from the corresponding author.

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
