# Peer review of "Hybrid Collagen Hydrogel/Chondroitin-4-Sulphate Fortified with Dermal Fibroblast Conditioned Medium for Skin Therapeutic Application"

_polymers, 2021, doi:10.3390/polym13040508_

Round 1
Reviewer 1 Report
The hybrid biomaterials based on collagen hydrogel, chondroitin and dermal fibroblast conditioned medium might be of interest to develop skin therapeutic application.
Adequate techniques for physical-chemical and morphological characterization of the new constructs are properly applied.
However, the Discussion section must be significantly improved. The first part, lines 200 - 300 should be reformulated, as some sentences have no verb, or need verb at singular, or are not clear. Other parts looks like a literature survey, or contain definitions (e.g. lines 358-359). Some unclear terms are used, such as crystalline formation (lines 332, 352) instead of crystalline phase formation, etc.
Moreover, the text of the figures have to present the significance of images or graphics, without further explanations which are usually found in the paper text or should be added in the text.
At page 4 the units are not correctly expressed: line 143 - 1.5 mL, line 164 - 0.789 g/mL, line 190 - 200 µL. Sentence at lines 242-243 - should be reformulated. Actually, the entire text should be carefully corrected for editing of English.
Finally, the conclusions do not reflect the scientific content of the paper and should be completed.
Author Response
RESPONSE TO REVIEWER’S COMMENTS
Manuscript ID: polymers-1091070
Type of manuscript: Article
Title: Hybrid Collagen Hydrogel/Chondroitin-4-Sulphate Fortified with Dermal Fibroblast Conditioned Medium for Skin Therapeutic Application
Dear Editor/ Reviewers,
Thank you very much for the valuable comments and suggestions. We have made changes according to the given recommendations and the manuscript was sent for proofreading and English editing. Following are the point-to-point responses to the comments. All changes were highlighted in the text.
Reviewer(s)' Comments to Author:
Reviewer 1
The hybrid biomaterials based on collagen hydrogel, chondroitin and dermal fibroblast conditioned medium might be of interest to develop skin therapeutic application.
Adequate techniques for physical-chemical and morphological characterization of the new constructs are properly applied.
- However, the Discussion section must be significantly improved. The first part, lines 200 - 300 should be reformulated, as some sentences have no verb, or need verb at singular, or are not clear. Other parts looks like a literature survey, or contain definitions (e.g. lines 358-359). Some unclear terms are used, such as crystalline formation (lines 332, 352) instead of crystalline phase formation, etc.
Thank you for your comment. The sentences in the results and discussion section have been revised and the manuscript was sent for proofreading and English editing.
- Moreover, the text of the figures have to present the significance of images or graphics, without further explanations which are usually found in the paper text or should be added in the text.
Thank you for your comment. The figure legends have been revised and explained the figures clearly with necessary information and significant values (Figures 1-5).
- At page 4 the units are not correctly expressed: line 143 - 1.5 mL, line 164 - 0.789 g/mL, line 190 - 200 µL.
Thank you for your comment. The units for mL and µL already revised accordingly in materials and method section (page 4, line 141, 161 and page 5, line 187).
- Sentence at lines 242-243 - should be reformulated.
The sentences at line 242-243 have been revised to “Most of the constructs showed collagen fibril aggregates, resulting in thicker collagen fibrils. The homogeneously overlapping collagen fibrils observed in Col/C4S/DFCM-FM had fewer collagen fibrils arranged in bundles (page 8 line 236-238).
- Actually, the entire text should be carefully corrected for editing of English.
The manuscript was sent for proofreading and English editing.
- Finally, the conclusions do not reflect the scientific content of the paper and should be completed.
The conclusion already revised accordingly to reflect the objective of the study to “Our findings show that the fabricated Col/C4S/DFCM hydrogel constructs maintain the characteristics of collagen. The swelling and degradation properties of the constructs are suitable for the diffusion of DFCM, and the construct could have potential as a readily available acellular skin substitute that acts as an alternative carrier for delivering essential mediators and growth factors that enhance wound healing. We will investigate the efficacy of this delivery system in enhancing skin regeneration and its therapeutic application in future studies.” (Conclusion section, page 12, line 383-389).

Reviewer 2 Report
Overall it's an interesting paper, thoughtful, properly designed and described. It only requires some minor adjustments as indicated below:
- Celsius degree symbol for improvement in many places in the manuscript
- What solvent was used to prepare the collagen and chondroitin-4-sulphate solution?
- Porosity of Col/C4S/DFCM Constructs/Swelling Analysis - with what equipment was the lyophilization carried out?
- Figure 2 - too many significant figures in the results describing the position of the bands, 4 significant figures would be sufficient. Some numbers overlap.
Author Response
RESPONSE TO REVIEWER’S COMMENTS
Manuscript ID: polymers-1091070
Type of manuscript: Article
Title: Hybrid Collagen Hydrogel/Chondroitin-4-Sulphate Fortified with Dermal Fibroblast Conditioned Medium for Skin Therapeutic Application
Dear Editor/ Reviewers,
Thank you very much for the valuable comments and suggestions. We have made changes according to the given recommendations and the manuscript was sent for proofreading and English editing. Following are the point-to-point responses to the comments. All changes were highlighted in the text.
Reviewer(s)' Comments to Author:
Reviewer 2
Overall it's an interesting paper, thoughtful, properly designed and described. It only requires some minor adjustments as indicated below:
- Celsius degree symbol for improvement in many places in the manuscript
Thank you for your comment. All Celsius degree symbols already revised accordingly in the text manuscript.
- What solvent was used to prepare the collagen and chondroitin-4-sulphate solution?
The 0.35M acetic acid was used as a solvent for the collagen hydrogel whilst the chondroitin-4-sulphate is in a powder form and was mixed directly to the collagen hydrogel with concentration of 2.4 % (w/v). The collagen hydrogel with C4S was neutralized by adding 1 M sodium hydroxide (NaOH; Sigma) until pH 7.
Porosity of Col/C4S/DFCM Constructs/Swelling Analysis - with what equipment was the lyophilization carried out?
All collagen constructs were lyophilized or freeze-dried overnight using a freeze dryer (Ilshin, korea) (method section, page 4, line 154,164-165).
- Figure 2 - too many significant figures in the results describing the position of the bands, 4 significant figures would be sufficient. Some numbers overlap
Thank you for your comment. However, figure 2 is the IR spectrum results that represents the five different constructs in this study. We hope we can proceed with the five IR spectrum in figure 2, with different colour and no numbers overlap. (Figure 2, page 7)

Reviewer 3 Report
The authors report fabrication of hybrid collagen hydrogel/chondroitin-4-sulphate fortified with dermal fibroblast conditioned medium, which is potential for skin therapeutic application. The work is systematical and interesting. However, some problems present. I suggest a major revision of the manuscript before acceptance for publication.
Detailed comments and suggestions are as follows.
1. Writing problems. Examples: (1) "The combination of biomaterial and cells secreted protein or biomolecules would be the 15 current strategy for rapid treatment of wound healing."---"The combination of biomaterial and proteins or biomolecules secreted by cells is the current strategy for rapid wound healing treatment." (2) Page 6, line 217, "The IR spectrum from FTIR analysis showed peak absorbance that represents the chemical composition of the collagen (Figure 2)."---"The IR spectrum showed peak absorbance that represents the chemical composition of collagen (Figure 2)." Similar problems in many other places of the manusctipt and the authors should revise thoroughly the whole manuscript.
2. I suggest the authors revising the ABSTRACT section by removing the experimental details from it so that the section can focus on the precise results and conclusions.
3. Figure 2, the x axis, "Absorption"---"Wavenumber".
4. Figure 3, in some images, for example, 5000X of Col/C4S/DFCM-KM2, some crystal-like aggretates can be found. What are these aggregates?
5. Several figures can be reformatted to be more beautiful.
6. More related references can be cited in the manuscript, for example, "Polymers 2020, 12(3), 580".
Author Response
RESPONSE TO REVIEWER’S COMMENTS
Manuscript ID: polymers-1091070
Type of manuscript: Article
Title: Hybrid Collagen Hydrogel/Chondroitin-4-Sulphate Fortified with Dermal Fibroblast Conditioned Medium for Skin Therapeutic Application
Dear Editor/ Reviewers,
Thank you very much for the valuable comments and suggestions. We have made changes according to the given recommendations and the manuscript was sent for proofreading and English editing. Following are the point-to-point responses to the comments. All changes were highlighted in the text.
Reviewer(s)' Comments to Author:
Reviewer 3
The authors report fabrication of hybrid collagen hydrogel/chondroitin-4-sulphate fortified with dermal fibroblast conditioned medium, which is potential for skin therapeutic application. The work is systematical and interesting. However, some problems present. I suggest a major revision of the manuscript before acceptance for publication.
Detailed comments and suggestions are as follows.
Writing problems. Examples: (1) "The combination of biomaterial and cells secreted protein or biomolecules would be the current strategy for rapid treatment of wound healing
Thank you for your comment. The sentence in the abstract was revised to “The current strategy for rapid wound healing treatment involves combining a biomaterial and cell-secreted proteins or biomolecules”. (Abstract, page 1, line 15-16).
Page 6, line 217, "The IR spectrum from FTIR analysis showed peak absorbance that represents the chemical composition of the collagen (Figure 2).
The sentence has been revised to “The IR spectra showed peak absorbance that represented the chemical composition of the constructs (Figure 2)”(page 6, line 213-214).
- Similar problems in many other places of the manuscript and the authors should revise thoroughly the whole manuscript.
The manuscript was sent for proofreading and English editing.
- I suggest the authors revising the ABSTRACT section by removing the experimental details from it so that the section can focus on the precise results and conclusions.
We agree with the reviewer’s comment. However, the experimental detail is really important to the reader to better understand the study design and findings of this study.
- Figure 2, the x axis, "Absorption"---"Wavenumber".
The x-axis in figure 2 has been changed to “Wavenumber (cm-1)” (Figure 2, page 7).
Figure 3, in some images, for example, 5000X of Col/C4S/DFCM-KM2, some crystal-like aggregates can be found. What are these aggregates?
There is crystal like aggregates in most of the constructs in figure 3 might be due to the crystalline phase during freeze-drying closing the pores of the constructs or because of the presence of a small amount of salts in the DFCM, which usually tends to hydrolyse in water and form discrete nanoparticles [48]. (Discussion section, page 11, line 324-328).
Several figures can be reformatted to be more beautiful.
Thank you for your comment. Figure 2, 4 and 5 already rearrange and reformatted accordingly. (Figure 2, 4, 5)
More related references can be cited in the manuscript, for example, "Polymers 2020, 12(3), 580".
Thank you for your suggestion. A few related references have been cited in the discussion section of the manuscript (page 11, line 296-297, 329-334).
- Elkhoury, K.; Russell, C.S.; Sanchez-Gonzalez, L.; Mostafavi, A.; Williams, T.J.; Kahn, C.; Peppas, N.A.; Arab-Tehrany, E.; Tamayol, A. Soft-Nanoparticle Functionalization of Natural Hydrogels for Tissue Engineering Applications. Advanced Healthcare Materials 2019, 8, 1900506, doi:https://doi.org/10.1002/adhm.201900506.
- Xu, X.; Liu, Y.; Fu, W.; Yao, M.; Ding, Z.; Xuan, J.; Li, D.; Wang, S.; Xia, Y.; Cao, M. Poly(N-isopropylacrylamide)-Based Thermoresponsive Composite Hydrogels for Biomedical Applications. Polymers 2020, 12, 580.
- Haidari, H.; Kopecki, Z.; Sutton, A.T.; Garg, S.; Cowin, A.J.; Vasilev, K. pH-Responsive “Smart” Hydrogel for Controlled Delivery of Silver Nanoparticles to Infected Wounds. Antibiotics 2021, 10, 49.

Round 2
Reviewer 1 Report
The authors followed the recommendations and made corrections accordingly.